# Structural, Magnetic, and Magneto-Thermal Properties of Rare Earth Intermetallic GdRhIn

**DOI:** 10.3390/s24196326

**Published:** 2024-09-30

**Authors:** Ravinder Kumar, Arrab Ali Maz, Satyendra Kumar Mishra, Sachin Gupta

**Affiliations:** 1Department of Physics, Bennett University, Greater Noida 201310, India; 2Space and Resilient Communications and Systems (SRCOM), Center Technologic de Telecomunicacions de Catalunya (CTTC), Avinguda Carl Friedrich Gauss, 11, 08860 Castelldefels, Spain; 3Centre of Excellence in Nanosensors and Nanomedicine, Bennett University, Greater Noida 201310, India

**Keywords:** magnetocaloric effect, GdRhIn, rare earth intermetallic, magnetization

## Abstract

We study the structural, magnetic, and magneto-thermal properties of the GdRhIn compound. The room-temperature X-ray diffraction measurements show a hexagonal crystal structure. Temperature and field dependence of magnetization suggest two magnetic transitions—antiferromagnetic to ferromagnetic at 16 K and ferromagnetic to paramagnetic at 34 K. The heat capacity measurements confirm both the magnetic transitions in GdRhIn. The magnetization data were used to calculate isothermal magnetic entropy change and refrigerant capacity in GdRhIn, which was found to be 10.3 J/Kg-K for the field change of 70 kOe and 282 J/Kg for the field change of 50 kOe, respectively. The large magnetocaloric effect in GdRhIn suggests that the material could be used for magnetic refrigeration at low temperatures.

## 1. Introduction

Magnetism originating from unpaired 4*f* shell electrons in rare earths makes them interesting for fundamental physics as well as for application purposes. When these materials are combined with the transition metals, they show outstanding properties emerging from rare earth and transition metal sublattices. Rare earth intermetallics are being investigated for various applications such as permanent magnets, magnetic sensors, magnetic refrigeration, biomedical, and spintronics [1,2,3,4,5,6,7,8]. Depending on their magnetic and electronic properties, these materials are explored for various types of sensors, such as magneto-resistive sensors, quantum sensors, Hall sensors, and chemical sensors [9,10,11,12,13,14]. Among rare earth intermetallics, *RTX* (*R* = rare earth, *T* = transition metal, *X* = main group element) compounds were found to show many unusual thermal and magnetic properties, which have drawn significant attention from scientists and engineers for their prominence in sensor applications [1,2,4]. Gupta and Suresh studied the physical properties of various *RTX* compounds in detail to correlate their structural, magnetic, and magnetocaloric properties, as referenced in Ref. [1]. It has been observed that these materials could be promising for their applications in magnetic sensors and magnetic refrigeration [1,2,4]. Magnetic refrigeration is a new technology that is environmentally friendly, highly efficient, and has low power consumption [15]. Magnetic refrigeration is based on the magnetocaloric effect (MCE), which is an intrinsic property of magnetic materials, allowing materials to heat up or cool down subjected to their magnetization and demagnetization [16]. To replace conventional gas compression-based refrigeration with magnetic refrigeration technology, we need to explore materials with novel properties, such as low hysteresis and strong mechanical and chemical stability [15,17,18,19]. Several Gadolinium (Gd)-based intermetallics have been studied for magnetic refrigeration and were found to show excellent properties [5,20,21,22,23]. Motivated by these results, we investigate the magnetic and related properties of GdRhIn, which belongs to the *R*RhIn family. *R*RhIn has been reported to crystallize in two crystal structures—*R*RhIn with *R* = La-Nd, Sm, Gd-Tm in ZrNiAl type hexagonal and *R* = Eu, Yb, Lu in TiNiSi type orthorhombic crystal structure, respectively [24,25,26,27,28]. EuRhIn and GdRhIn in this series were found to show ferromagnetic (FM) behavior below 22 and 34 K, respectively [29,30]. Though most of the members in this series lack detailed study on magnetic and related properties, in this paper, we discuss combined structural, magnetic, and magneto-thermal properties of the GdRhIn compound. We synthesized the GdRhIn compound using the arc melt technique and studied structural, magnetic, and magneto-thermal properties. Crystal structure refinement confirms hexagonal crystal structure in GdRhIn. The magnetic measurements show double magnetic transitions at low temperatures. To confirm it, we carried out the heat capacity measurements. To know the potential of this material for magnetic refrigeration applications, we calculated the magnetocaloric effect (MCE) from magnetization data and observed that GdRhIn shows a large MCE.

## 2. Materials and Methods

The arc melting technique was used to melt constituent elements Gd, Rh, and In. The atomic purity for each element was better than 99.9%. The constituent elements (Gd, Rh, In) were taken in a 1:1:1 atomic percentage ratio. The current applied to melt the alloy was in the range of 150–200 A, and the sample was melted for a few minutes. To ensure better homogeneity, the formed ingot was flipped multiple times and re-melted. The arc-melted sample was sealed in an evacuated (10^−6^ torr) quartz tube and annealed in a furnace for 7 days at 800 °C. The X-ray diffraction measurement of the powder sample was performed at the X’PERT PRO diffractometer using Cu Kα radiation in the 2θ range of 20–80 degrees with a step size of 0.017 degrees at room temperature. Magnetic measurements were performed at a Vibrating Sample Magnetometer (VSM) attached to a Physical Property Measurement System (Quantum Design, model −6500). The same PPMS model was also used to measure the heat capacity of the sample using the thermal relaxation method. A total of 5 mg sample of GdRhIn was placed in the calorimeter. The sample temperature was monitored during the heating and cooling process, and the raw data were fitted with the “two – τ” model. The XRD and magnetic measurements were repeated to confirm the reproducibility of the sample.

## 3. Results and Discussion

### 3.1. Crystal Structure

Figure 1 shows the powder XRD pattern recorded at room temperature for the GdRhIn compound. The obtained XRD pattern was analyzed with Rietveld refinement using FullProf Suite software [31]. The Rietveld refinement of the XRD pattern confirms that GdRhIn crystallizes in a Fe_2_P-type hexagonal crystal structure with space group P-62m (space group #189). The lattice parameters obtained from the refinement are *a* = *b* = 7.53 Å and *c* = 3.92 Å for the GdRhIn compound.

### 3.2. Magnetic Properties

The temperature, *T* and magnetic field, *H* dependence of magnetization, and *M* for GdRhIn are shown in Figure 2. Figure 2a shows the temperature dependence of magnetization under zero field-cooled (ZFC) and field-cooled (FC) configurations at an applied field of 500 Oe. The difference between ZFC and FC modes is that, in the first case, the sample is cooled in the absence of any magnetic field, while in the latter case, it is cooled in the presence of a magnetic field. The magnetization in both cases is recorded in the presence of a magnetic field as the sample is heated. It can be noted from Figure 2a that on decreasing the temperature, the GdRhIn shows a paramagnetic to ferromagnetic transition at 34 K. At 16 K, the magnetization shows a cusp, suggesting an antiferromagnetic transition [8,32,33]. A similar low-temperature behavior was also observed in TbRhSn and GdRhGe compounds [34]. At low temperatures, the difference in magnetization for ZFC and FC is attributed to thermomagnetic irreversibility, as seen in many *RTX* compounds (please see the inset). The Curie–Weiss fit to susceptibility data yield the values of paramagnetic Curie temperature (*θ*_p_) and effective magnetic moment (*µ*_eff_) to be 24.3 K and 8 µ_B_/Gd^3+^_,_ respectively. It is worth noting that the *θ*_p_ value obtained with a positive sign suggests that ferromagnetic correlations among the magnetic moments are strong. The observed value of *µ*_eff_ is very close to the theoretical expected value of 7.94 µ_B_/Gd^3+^ for the Gadolinium ion. Figure 2b shows the field dependence of magnetization measured at different temperatures. At the field of 90 kOe, the magnetization in GdRhIn shows clear saturation. It can be noted that at low magnetic fields, the magnetization increases with increasing temperature up to 16 K, indicating an antiferromagnetic transition. At higher magnetic fields, the magnetization decreases with temperature, indicating a ferromagnetic transition below 34 K. The saturation moment for a field value of 90 kOe and at 5 K was found to be 6.9 µ_B_, which is very close to the theoretical value of 7 µ_B_/Gd^3+^. Strong temperature dependence of magnetization near Curie temperature suggests that GdRhIn could be a promising candidate for magnetic refrigeration and temperature-sensitive sensor applications.

### 3.3. Heat Capacity

In order to know more about the thermal and magnetic behavior of GdRhIn, the heat capacity, *C* measurement was performed at zero magnetic field, as shown in Figure 3. The temperature dependence of the heat capacity shows two anomalies corresponding to the two magnetic transitions as observed in magnetization data. The *λ*-type peak is observed around 16 K due to an antiferromagnetic transition, while the slight change in the slope can be seen around 34 K, which corresponds to a ferromagnetic transition in GdRhIn. The small *λ*-type peak corresponding to magnetic transition suggests a second-order type transition in GdRhIn [35]. The temperature dependence of the heat capacity (lattice and electronic) in a low-temperature regime can be described by the following equation:(1)$$C=\gamma T+\beta T^{3}$$
where *γ* and *β* are the coefficients of electronic heat capacity and thermal expansion, respectively. The Debye temperature, *θ*_D,_ can be deduced from the following relation [36]:(2)$$\theta_{D}=\sqrt[{3}]{12\pi^{4}R/5\beta }≅\sqrt[{3}]{1944/\beta }$$

The inset in Figure 3 shows a fit of Equation (1) to experimental heat capacity data at low temperatures. The *γ* and *β* values estimated from the fit are found to be 0.79 and 0.0006, respectively. Due to the low ordering temperature in GdRhIn, the value of *γ* could be deviated from a reasonable value. The Debye temperature, *θ*_D,_ estimated using Equation (2), is found to be 148 K, which is close to values observed in other similar *RTX* compounds [37,38].

### 3.4. Magnetocaloric Effect (MCE)

The magnetocaloric effect (MCE) is an intrinsic property of magnetic material, which makes material heat up or cool down on the application or removal of the magnetic field. MCE can be measured in terms of change in isothermal magnetic entropy, ∆*S*_M_, and/or adiabatic temperature, ∆*T*_ad_. To know the potential of GdRhIn for magnetic refrigeration applications, the change in the isothermal magnetic entropy, ∆*S*_M_, was calculated from the magnetization, *M* (*H*, *T*) data using the following Maxwell’s relation [5]:(3)$$\Delta S_{M}={\int_{0}^{H}\left[\frac{\partial M}{\partial T}\right]}_{H}dH$$

The above relation was simplified in the following equation, which can be used for calculating isothermal magnetic entropy change:(4)$$\Delta S_{M}=\sum_{i}\frac{M_{i+1}-M_{i}}{T_{i+1}-T_{i}}\Delta H_{i}$$
where *M_i_*_+1_ and *M_i_* represent the magnetization at temperatures *T_i_*_+1_ and *T_i_*, respectively, in the field H*_i_* [39].

Figure 4a shows the temperature dependence of Δ*S_M_* as a function of the magnetic field for the GdRhIn compound. It can be observed from the plot that the Δ*S_M_* increases with the field and shows a value of 10.3 J/Kg-K for the field change of 70 kOe. The obtained value of Δ*S_M_* in GdRhIn is comparable to or higher than several Gd-based or other rare earth intermetallics shown in Table 1. The temperature dependence of Δ*S_M_* shows a broad peak around Curie temperature, which is expected in ferromagnetic materials. A slight change in slope is observed at low temperatures, which may be attributed to the antiferromagnetic transition observed at 16 K in the magnetization data. Additional low-temperature data would provide further clarity about the MCE.

In addition to ∆*S_M_* and ∆*T_ad_*, another important parameter with respect to the magnetic cooling efficiency of materials is the refrigerant capacity (RC) or relative cooling power (RCP), which can be calculated as follows [45]:(5)$$RC=\int_{T_{1}}^{T_{2}}\Delta S_{M}(T)dT$$
where *T*_1_ and *T*_2_ represent the range of temperature at half maximum in Δ*S_M_*. In an ideal refrigeration cycle, RC is the measure of the amount of heat transferred from the cold to a hot reservoir. The RC values were calculated from the ∆*S_M_* vs. *T* plot and are shown in Figure 4b for GdRhIn. It exhibits a linear dependence on the magnetic field and is found to be 282 J/Kg for the field change of 50 kOe. RC could not be calculated for fields higher than 50 kOe due to a smaller number of data points. The value of RC is comparable to or higher than other Gd-based rare earth intermetallics, as shown in Table 1. The large magnetocaloric effect in GdRhIn could make this material a good candidate for magnetic refrigeration and temperature sensors. The magnetocaloric effect (MCE) in GdRhIn may be further explored by substituting other rare earth or transition metal elements. Such substitutions could offer a range of magnetic transition temperatures, thereby broadening the MCE peak and making the material promising for applications across a wider temperature range.

## 4. Conclusions

The GdRhIn was synthesized using the arc melt technique. The crystal structure analysis shows a hexagonal structure. GdRhIn shows two magnetic transitions at low temperatures. It shows antiferromagnetic to ferromagnetic transition at 16 K and ferromagnetic to paramagnetic transition at 34 K. Both the magnetic transitions are confirmed in the heat capacity data. The Debye temperature calculated from the heat capacity data is in range with other members of the RTX family. The magnetocaloric effect calculated from the magnetization data was found to be larger than the other Gd-based rare earth intermetallics, indicating GdRhIn to be a promising material for low-temperature magnetic refrigeration. Moreover, strong temperature-dependent magnetization, large heat capacity, and magnetocaloric effect suggest that GdRhIn could also be used for temperature-sensitive sensor applications.

## Figures and Tables

**Figure 1 sensors-24-06326-f001:**
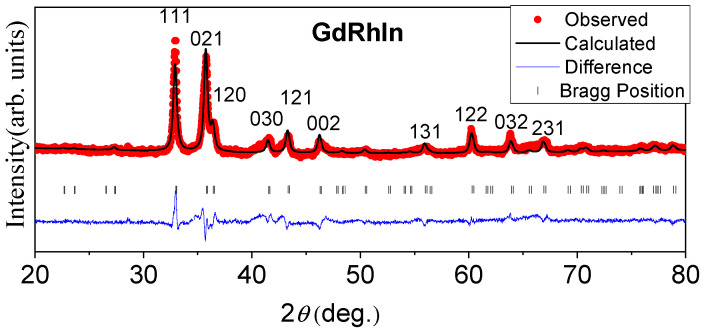
Powder X-ray diffraction pattern for GdRhIn compound. The bottom plot shows the difference between the theoretically calculated and experimentally observed data. The high-intensity peaks are indexed with the Bragg notation.

**Figure 2 sensors-24-06326-f002:**
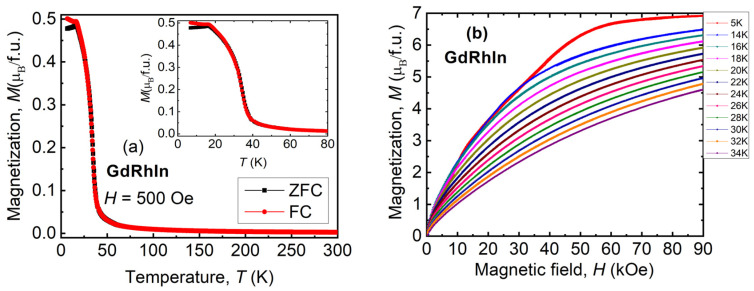
(**a**) The temperature, *T* dependence of magnetization, *M* at a magnetic field of 500 Oe recorded under zero field-cooled (ZFC) and field-cooled (FC) modes. (**b**) The field, *H* dependence of magnetization, was recorded at different temperatures for the GdRhIn compound. The inset shows a magnified view of the M vs. T plot at 500 Oe field. Symbols: µ_B_—Bohr Magneton, f.u.—formula unit.

**Figure 3 sensors-24-06326-f003:**
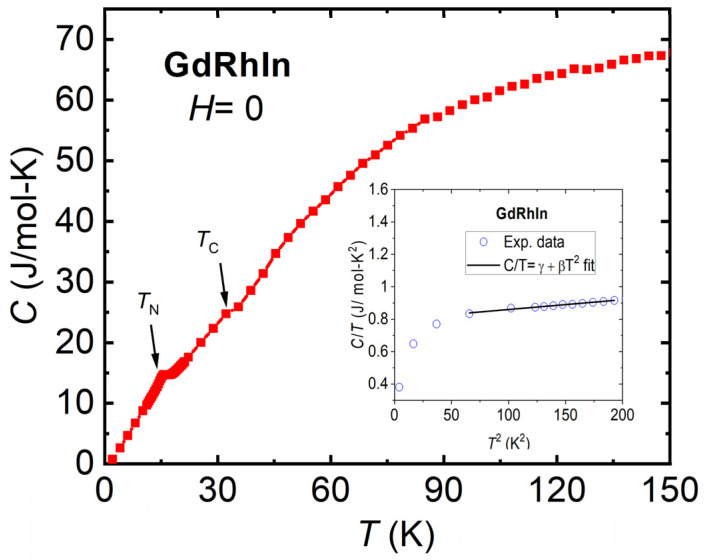
The temperature, *T* dependence of the heat capacity, C, measured at zero magnetic field, *H*. The arrows are a guide to magnetic transition temperatures (*T*_N_ and *T*_C_). The inset shows the fit to heat capacity data in a low-temperature regime.

**Figure 4 sensors-24-06326-f004:**
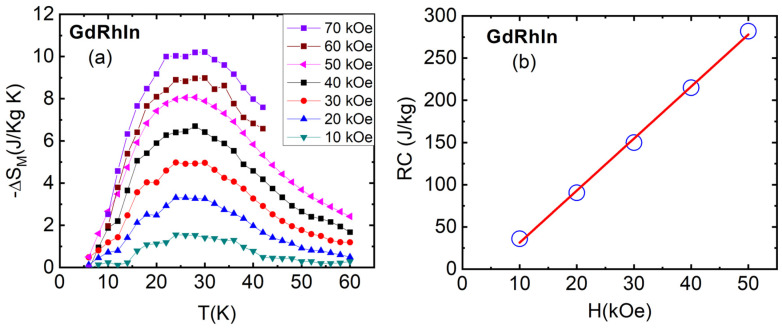
(**a**) The temperature dependence of isothermal magnetic entropy change, Δ*S_M_*_,_ as a function of magnetic field change, ∆*H* for GdRhIn. (**b**) The field dependence, *H*, of refrigerant capacity (RC) for GdRhIn compound.

**Table 1 sensors-24-06326-t001:** Isothermal magnetic entropy change, Δ*S_M_*_,_ and refrigerant capacity (RC) for different materials for the field change (∆*H*) of 50 kOe.

Material	Δ*S_M_* (J/kg K) at 50 kOe	RC (J/kg) at 50 kOe	Reference
GdScSi	3.7	133.2 *	[40]
GdScGe	4.0	135.3 *	[40]
GdRhSn	6.5	-	[41]
GdRhGe	~1	-	[34]
GdRuSi	10.7	336	[42]
Gd_3_Co_2_Ge_4_	7.9	184	[43]
Tb_3_Co_2_Ge_4_	3.6	70	[43]
Gd_5_Pt_2_In_4_	2.2	139	[44]
Tb_5_Pt_2_In_4_	1.7	165	[44]
GdRhIn	8	282	This Work

* for ∆*H* = 20 kOe.

## Data Availability

The datasets can be made available on request.

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
