# Peer review of "Structural, Magnetic, and Magneto-Thermal Properties of Rare Earth Intermetallic GdRhIn"

_sensors, 2024, doi:10.3390/s24196326_

Round 1

Reviewer 1 Report

Comments and Suggestions for Authors

This article presents a study on the structural, magnetic, and magneto-thermal properties of the GdRhIn compound, which possesses certain scientific significance and potential application value. The employed experimental methods, including X-ray diffraction, magnetic measurements, and heat capacity measurements, are reasonable and enable a comprehensive characterization of the compound. The results analysis is comparatively detailed, covering the discussion of magnetic transitions, heat capacity anomalies, and magnetocaloric effect, along with comparisons with other related compounds. The conclusion is clear, summarizing the key properties and potential applications of the GdRhIn compound. Overall, the article appears to be of acceptable quality. However, to make it more robust, the following suggestions could be considered:

1. Provide more background information on the significance and potential applications of GdRhIn in the introduction.

2. Add a more comprehensive review of the research status in this field to highlight the novelty and importance of this study.

3. Provide detailed information on the ratio of each element and the specific melting conditions during sample preparation, such as arc current and melting time.

4. Supplement the specific parameter settings for the X-ray diffraction measurement, such as the scanning range and step size.

5. Provide more information about the instrument models and measurement conditions for magnetic measurements and heat capacity measurements.

6. When discussing the magnetic transitions, more theoretical models or literature references can be used to explain the underlying physical mechanisms.

7. The analysis of the heat capacity measurement results can be more in-depth, exploring its relationship with the crystal structure and magnetic properties.

8. When presenting the magnetocaloric effect results, add comparative data with other known magnetic refrigeration materials to better demonstrate the advantages of GdRhIn.

9. The error analysis of the data can be more detailed, such as providing the error range for magnetic measurements and heat capacity measurements.

10. Enhance the discussion by comparing the properties of GdRhIn with those of similar compounds in more detail.

11. Further prospects for the challenges and future research directions of this material in practical applications can be included.

12. Optimize the quality and resolution of the figures and tables to ensure clear and readable data.

13. Explain the symbols and abbreviations in the figures and tables to make it easier for readers to understand.

14. Ensure the format of the references is uniform and accurate.

15. Supplement some recent relevant research literature to showcase the latest progress in this field.

16. Discuss the relationship between the performance of the GdRhIn compound and its structure more deeply, and how to optimize its performance through structural regulation.

17. Compare and analyze the performance of this compound with other similar compounds to explore its advantages and disadvantages.

18. Add information about the repeatability and reliability of the experiments, such as conducting multiple experiments and providing statistical analysis of the data.

19. Add topic sentences at the beginning of each paragraph to help readers better understand the content of the paragraph.

Comments on the Quality of English Language The quality of the English language in this article is generally good. The writing is clear and understandable, and the technical terms are used appropriately. However, there are a few areas that could be improved:

  1. There are some minor grammatical errors and typos, such as missing articles or incorrect verb tenses. A careful proofreading would be helpful to eliminate these errors.
  2. The language could be more precise in some places. For example, when describing the experimental results, more specific and detailed language would make the findings more clear and convincing.
  3. The phrasing could be more varied to enhance the readability of the article. Using a wider range of vocabulary and sentence structures would make the text more engaging.

Author Response

Attached the response of comments

Reviewer 2 Report

Comments and Suggestions for Authors

Fig. 2a is not informative. Scale was chosen in not very good manner. Authors are recommended to replace this figure or add some insert showing temperature range of 0-80 deg. in more detailed form.

Fig. 2b. It is not clear, what is the difference between the large plot and insert. Both show M vs. H plots in different temperatures. Is this magnification really necessary?

Discussion part is not obvious. Authors should compare obtained results with data published by other researchers.

There are only 6 references published during last five years. The list of references should be actualized.

Comments on the Quality of English Language

There are some misprints and mistakes

Author Response

Attached the response of comments

Round 2

Reviewer 2 Report

Comments and Suggestions for Authors

Manuscript was improved and can be published

Author Response

Comment: Manuscript was improved and can be published.

Response: We thank the referee for his/her valuable time in going through the revised manuscript and recommending its publication.
